# Low-Dimensional Architectures in Isomeric *cis*-PtCl_2_{Ph_2_PCH_2_N(Ar)CH_2_PPh_2_} Complexes Using Regioselective-N(Aryl)-Group Manipulation

**DOI:** 10.3390/molecules26226809

**Published:** 2021-11-11

**Authors:** Peter De’Ath, Mark R. J. Elsegood, Noelia M. Sanchez-Ballester, Martin B. Smith

**Affiliations:** Department of Chemistry, Loughborough University, Loughborough LE11 3TU, UK; P.DeAth@lboro.ac.uk (P.D.); m.r.j.elsegood@lboro.ac.uk (M.R.J.E.); n.m.sanchez-ballester@lboro.ac.uk (N.M.S.-B.)

**Keywords:** amide groups, isomers, late-transition metals, P-ligands, phenols, secondary interactions, single crystal X-ray crystallography

## Abstract

The solid-state behaviour of two series of isomeric, phenol-substituted, aminomethylphosphines, as the free ligands and bound to Pt^II^, have been extensively studied using single crystal X-ray crystallography. In the first library, isomeric diphosphines of the type Ph_2_PCH_2_N(Ar)CH_2_PPh_2_ [**1a**–**e**; Ar = C_6_H_3_(Me)(OH)] and, in the second library, amide-functionalised, isomeric ligands Ph_2_PCH_2_N{CH_2_C(O)NH(Ar)}CH_2_PPh_2_ [**2a**–**e**; Ar = C_6_H_3_(Me)(OH)], were synthesised by reaction of Ph_2_PCH_2_OH and the appropriate amine in CH_3_OH, and isolated as colourless solids or oils in good yield. The non-methyl, substituted diphosphines Ph_2_PCH_2_N{CH_2_C(O)NH(Ar)}CH_2_PPh_2_ [**2f**, Ar = 3-C_6_H_4_(OH); **2g**, Ar = 4-C_6_H_4_(OH)] and Ph_2_PCH_2_N(Ar)CH_2_PPh_2_ [**3**, Ar = 3-C_6_H_4_(OH)] were also prepared for comparative purposes. Reactions of **1a**–**e**, **2a**–**g**, or **3** with PtCl_2_(η^4^-cod) afforded the corresponding square-planar complexes **4a**–**e**, **5a**–**g**, and **6** in good to high isolated yields. All new compounds were characterised using a range of spectroscopic (^1^H, ^31^P{^1^H}, FT–IR) and analytical techniques. Single crystal X-ray structures have been determined for **1a**, **1b**∙CH_3_OH, **2f**∙CH_3_OH, **2g**, **3**, **4b**∙(CH_3_)_2_SO, **4c**∙CHCl_3_, **4d**∙½Et_2_O, **4e**∙½CHCl_3_∙½CH_3_OH, **5a**∙½Et_2_O, **5b**, **5c**∙¼H_2_O, **5d**∙Et_2_O, and **6**∙(CH_3_)_2_SO. The free phenolic group in **1b**∙CH_3_OH, **2f**∙CH_3_OH_,_
**2g**, **4b**∙(CH_3_)_2_SO, **5a**∙½Et_2_O, **5c**∙¼H_2_O, and **6**∙(CH_3_)_2_SO exhibits various intra- or intermolecular O–H∙∙∙X (X = O, N, P, Cl) hydrogen contacts leading to different packing arrangements.

## 1. Introduction

Tertiary phosphines, and their phosphine oxides, have played an important role in the study of supramolecular and self-assembly processes [1,2,3]. Their synthetic versatility, coupled with ease of substituent modification, has no doubt played a significant contribution over the years. Hydrogen bonding interactions are routinely encountered in supramolecular ligand systems as illustrated by the elegant studies from Breit [4], Reek [5], and others [6,7]. More recently, amongst other common types of non-covalent interactions, those based on halogen bonding [8,9] and H^δ+^∙∙∙H^δ−^ have been reported [10]. 

For a number of years, we [11,12,13,14,15,16], and others [17,18,19,20,21,22], have been interested in aminomethylphosphines, readily amenable by Mannich condensation reactions. Such interest stems from the relative ease of accessing *P*-monodentate ligands based on a P–C–N linker [11,15,16,19,20,22] or *P*/*P*-bidentate derivatives bearing a P–C–N–C–P backbone [12,13,14,17,18,19,21]. Previously, we have shown that the N-arene group can be easily tuned with, for example, various H-bonding donor/acceptor sites based on –CO_2_H/OH groups [12,13,14,15,16]. In continuation of these studies, we report here our work on the regioselective positioning of amide/hydroxy and methyl groups within a series of aminomethylphosphines, both as the free ligands and when coordinated to a square-planar Pt(II) metal centre. Our rationale for introducing an –C(O)NH– group is based on the known use of this functionality in supramolecular chemistry [23] and, furthermore, the recent interest in amide-modified phosphines for their variable coordination chemistry [24,25,26], binding nitroaromatics [27], and relevance to catalysis based on Pd [28]. Our choice of metal fragment in this work, “*cis*-PtCl_2_”, is based on its capability to support a relatively small bite angle diphosphine ligand in a *cis*, six-membered ring conformation, and to provide up to two “acceptor” sites for potential H-bonding [29]. For this purpose, we elected to pursue a double Mannich condensation reaction of Ph_2_PCH_2_OH with a series of isomeric primary amines bearing either OH/CH_3_ groups and/or an amide spacer between the arene and P–C–N–C–P backbone ([Fig molecules-26-06809-ch001]).

## 2. Results and Discussion

### 2.1. Ligand Synthesis

We [11,12,13,14,15,16,29], and others [17,19,20,21,22], have previously used Mannich condensations as a versatile method for the synthesis of aminomethylphosphines. Accordingly, two equivalents of Ph_2_PCH_2_OH were reacted with one equivalent of the amine, for 24 h at r.t. under N_2_, yielding the desired phenol-substituted ditertiary phosphines **1a**–**e** and **3** (Figure 1).

For **1a**–**e**, colourless solids were isolated in 38–97% yields and found to be air stable in the solid state, but oxidise rapidly in solution. Compounds **1a**–**e** and **3** exhibit single resonances in their ^31^P{^1^H} NMR spectra (in d^6^-dmso) around δ(P) −26 ppm [12,13,14,15,29], indicating the presence of only one P^III^ environment. The ligands were also characterised by ^1^H NMR, FT–IR, and elemental analysis (Table 1). In particular, the absence of an NH resonance, in the ^1^H NMR spectra, confirmed that double condensation had occurred.

The synthesis of ditertiary phosphines, containing a flexible backbone presenting extra donor/acceptor sites with additional H-bonding capability, is described here with the opportunity to enhance solid-state packing behaviour. The precursors for the synthesis of the desired functionalised ditertiary phosphines **2a**–**g** were prepared using, in step (i), 1 equiv. of primary amine, *N*-carbobenzyloxyglycine (1 equiv.) and dicyclohexylcarbodiimide (DCC, 1 equiv.) in THF affording the corresponding carbamates followed by, in step (ii), treatment with Pd/C and cyclohexene in C_2_H_5_OH, to give the desired primary alkylamines in moderate to good yields [30,31]. Using a similar procedure to that described for **1a**–**e**, the amide-functionalised diphosphines **2a**–**e** were prepared in 65–89% yields by condensation using 1 equiv. of primary amine and two equiv. of Ph_2_PCH_2_OH at r.t. in CH_3_OH (Figure 1). Furthermore, the phenol-substituted phosphines **2f** and **2g** were synthesised to investigate what effect, if any, an absent methyl group on the N-arene ring displays. In the case of **2d**–**g**, the diphosphines were obtained as solids whereas **2a**–**c** were obtained as yellow oils that were sufficiently pure to be used in complexation studies. All compounds displayed a single ^31^P NMR resonance around δ(P) −26 ppm [12,13,14,15,29] indicating the inclusion of an amide spacer has negligible effect on the ^31^P chemical shift. Other spectroscopic and analytical data are given in Table 1.

### 2.2. Single Crystal X-ray Studies of ***1a***, ***1b***∙CH_3_OH_,_
***2f***∙CH_3_OH, ***2g***, and ***3***

X-ray quality crystals of **1a**, **1b**∙CH_3_OH_,_
**2f**∙CH_3_OH, **2g**, and **3** were obtained by slow evaporation of a methanol solution, while for **2g** diethyl ether was diffused into a deuterochloroform/methanol solution (Table 2). 

The geometry around each phosphorus atom is essentially pyramidal as would be anticipated (Figure 1, Figure 2, Figure 3, Figure 4 and Figure 5). The P^III^ atoms are in an *anti* conformation, presumably to minimise steric repulsions between the phenyl groups. The geometry about the N(1) centre is approx. pyramidal [Σ(C–N(1)–C) angles: 337.0(3)° for **1a**; 335(2)° for **1b**∙CH_3_OH; 335.2(2)/336.6(2)° for **2f**∙CH_3_OH; 333.7(2)° for **2g**] and approximately trigonal planar for **3** [Σ(C–N–C) = 359.05(11)°]. In **1a** and **1b**∙CH_3_OH, the N-arene ring [C(3) > C(8)] is twisted by ca. 88° (**1a**) and 86° (**1b**∙CH_3_OH) [12,32] such that it is almost perpendicular to the C(1)–N(1)–C(2) plane, whereas for **3**, the twist of the C(1)–N(1)–C(2) fragment is around 9° from co-planarity with the N-arene group, apparently as a result of the intermolecular H-bonding requirements (*vide infra*).

### 2.3. Secondary Interactions in ***1a***, ***1b***∙CH_3_OH, ***2f***∙CH_3_OH, ***2g***, and ***3***

The synthons observed in the solid state for these highly modular ligands may be dictated by various factors including the nature of the ligand, the flexibility of the P–C–N–C–P backbone, the predisposition of the OH/CH_3_ groups about the N-arene ring, and the solvent used in the crystallisation. In order to probe the OH/CH_3_ interplay of groups, the crystal structure of **1a**, with the –OH group in the *ortho* position with respect to the N(1) atom, is described first. Ligand **1a** crystallises with an intramolecular *S*(5) [33,34,35] H-bonded ring with *d* = 2.26(5) Å [denoting the hydrogen (H) to acceptor (A) distance in an H-bond D–H···A] [36] for the O–H···N interaction (Figure 1). The intramolecular H-bonding in **1a** limits the dimensionality of the packing of the diphosphine ligand. Therefore, the structure of **1a** is essentially zero-dimensional (Table 3). 

Compound **3**, where the −OH functional group is in the *meta* position with respect to the tertiary N(1) atom, aggregates in the solid state in such a way that fairly weak hydrogen bonds, O−H···P [*d* = 2.60(2) Å], form between symmetry-related molecules, creating dimers in which two ligands are held in an *R*^2^_2_(16) H-bonding motif (Figure 2). The distance between symmetry-related nitrogen atoms is 8.257 Å. The structure of **3** shows a 0D arrangement. 

Compound **1b**∙CH_3_OH, which contains the −OH group in a *para* position with respect to the N-arene, displays a similar structure to **3** with intramolecular O–H···P interactions at *d* = 2.60 Å. However, instead of forming dimers, there are 1D zig-zag chains in the *c* direction (Figure 3). The *para* hydroxyl oxygen acts as an acceptor for an O–H···O intermolecular H-bond from approximately alternate CH_3_OH molecules of crystallisation with *d* = 2.05 Å. These CH_3_OH molecules are 50/50 disordered with the second component H-bonding to its neighbour with *d* = 1.95 Å. Selected hydrogen parameters for **1b**∙CH_3_OH are listed in Table 3.

Compound **2f**∙CH_3_OH crystallises with two, similarly behaved, molecules in the asymmetric unit. A pair of H-bonded molecules, related by inversion symmetry, and with *d* = 1.81(3) Å for the intermolecular O–H···O interaction [1.78(3) Å for molecule 2] affords *R*^2^_2_(16) ring motifs (Figure 4). The intramolecular N–H∙∙∙N *S*(5) H-bond motif with *d* = 2.25(3) Å [2.26(3) Å for molecule 2] results in an intermediate twist angle of 64.23(13)° [but a rather more perpendicular 78.70(8)° for molecule 2] between planes C(1)/N(1)/C(2) and ring C(5) > C(10) [plane C(35)/N(4)/C(36) and ring C(39) > C(44) for molecule 2]. The *meta* hydroxy group in **2f** facilitates 0D dimer formation, as opposed to the chains observed in **2g** (*vida infra*).

For **2g**, molecules form H-bonded, 1D, zig-zag chains in the *c* direction via strong O–H∙∙∙O interactions with *d* = 1.83(5) Å (Figure 5). The intramolecular N–H∙∙∙N *S*(5), H-bond motif with *d* = 2.29(3) Å again results in an almost perpendicular twist angle of 82.09(15)° between planes C(1)/N(1)/C(2) and arene ring C(5) > C(10). The *para* hydroxy group promotes chain formation.

### 2.4. Dichloroplatinum(II) Complexes of ***1a***–***e***, ***2a***–***g***, and ***3***

The synthesis of *P*,*P*-chelate complexes *cis*-PtCl_2_(**1a**–**e**) [**4a**–**e**], *cis*-PtCl_2_(**2a**–**g**) [**5a**–**g**], and *cis*-PtCl_2_(**3**) [6] ([Fig molecules-26-06809-ch002]) was achieved by stirring the ligands and PtCl_2_(η^4^-cod) (1:1 ratio) in CH_2_Cl_2_ for 1.5 h with displacement of the cod ligand. The products were isolated in good yields as colourless solids. Downfield shifts of the ^31^P NMR resonances were observed for all complexes, with ^1^*J*_PtP_ coupling constants of approx. 3400 Hz, indicative of a *cis* conformation [29]. This was further supported by two characteristic ν_PtCl_ IR vibrations in the range of 279–316 cm^−1^ (Table 4). Furthermore, compounds **4a**–**e**, **5a**–**g**, and **6** present ν(NH) and ν(OH) IR absorptions in the range 3050–3465 cm^–1^ and also a strong band in the region of 1653–1675 cm^–1^, indicative of ν(C=O amide).

### 2.5. Single Crystal X-ray Studies of Complexes ***4b***∙(CH_3_)_2_SO, ***4c***∙CHCl_3_, ***4d***∙½Et_2_O, ***4e***∙½CHCl_3_∙½CH_3_OH, ***5a***∙½Et_2_O, ***5b***, ***5c***∙¼H_2_O, ***5d***∙Et_2_O, and ***6***∙(CH_3_)_2_SO

Detailed single crystal X-ray analysis (Table 5 and Table 6) of complexes **4b**∙(CH_3_)_2_SO, **4c**∙CHCl_3_, **4d**∙½Et_2_O, **4e**∙½CHCl_3_∙½CH_3_OH, **5a**∙½Et_2_O, **5b**, **5c**∙¼H_2_O, **5d**∙Et_2_O, and **6**∙(CH_3_)_2_SO shows that the geometry about each Pt(II) centre is approximately square planar [P–Pt–P range 90.23(9)–96.52(3)°] (Table 7 and Table 8). The Pt–Cl and Pt–P bond distances are consistent with literature values [29] and the conformation of the Pt–P–C–N–C–P six-membered ring in each complex is best described as a boat. The dihedral angle measured between the P_2_C_2_ plane and N-arene ring least-squares planes varies between 50.98(12)° [in **6**∙(CH_3_)_2_SO] and 90° (in **5d**∙Et_2_O), the difference of ca. 39° may tentatively be explained by the predisposition of the –OH group about the N-arene group and subsequent H-bonding requirements. Upon metal chelation, a degree of freedom, compared with the free ligands **1a**, **1b**∙CH_3_OH_,_
**2f**∙CH_3_OH, **2g**, and **3** has been removed, as the P–C–N–C–P backbone is locked into a specific conformation. Unfortunately, we were unable to obtain suitable X-ray quality crystals of compounds **4a** and **5e**–**g**.

Despite the *ortho* position of the hydroxy group in **4c**∙CHCl_3_, molecules do not form an intramolecular *S*(5) O–H∙∙∙N interaction as seen in **1a** (Figure 1), instead forming a bifurcated H-bond with the two coordinated chloride ligands of an adjacent molecule (Figure 6). This generates a 1D chain, and also attracts a bifurcated H-bonded chloroform molecule. There are somewhat asymmetric distances *d* for H(1C) to Cl(1) and Cl(2) are 2.45(4) and 2.76(4) Å, while those from H(34) to Cl(1) and Cl(2) are 2.66 and 2.86 Å, so are also asymmetric. The twist angle between planes P(1)/P(2)/C(1)/C(2) and ring C(3) > C(8) is 84.83(8)°, so is almost perpendicular. Atoms N(1) and Pt(1) lie 0.795(4) and 0.024(2) Å away from the P(1)/P(2)/C(1)/C(2) plane, respectively. The hinge angle across the P(1)–P(2) vector is 2.51(5)°. Selected hydrogen bonding geometric parameters for **4c**∙CHCl_3_ are shown in Table 9.

Compound **6**∙(CH_3_)_2_SO, in which the –OH group is *meta* to the N-arene group H-bonds to the DMSO molecule of crystallisation resulting in a 0D structure (Figure 7). The distance *d* for this H-bond is 1.79(2) Å. The twist angle between plane P(1)/P(2)/C(1)/C(2) and ring C(3) > C(8) is 50.98(12)°. Atoms N(1) and Pt(1) lie 0.758(4) and 0.404(2) Å away from the P(1)/P(2)/C(1)/C(2) plane, respectively, so is more chair-shaped than some of the other platinum(II) complexes reported here. The hinge angle across the P(1)–P(2) vector is 11.87(13)°.

For **4d**∙½Et_2_O (Figure 8) a molecule of badly disordered diethyl ether, modelled by the Platon Squeeze procedure, is not shown, but is in the vicinity of the hydroxy group and may H-bond to it resulting in a 0D structure. The twist angle between plane P(1)/P(2)/C(1)/C(2) and ring C(3) > C(8) is 67.82(7)°. Atoms N(1) and Pt(1) lie 0.797(3) and 0.2378(16) Å away from the P(1)/P(2)/C(1)/C(2) plane, respectively. The hinge angle across the P(1)–P(2) vector is 9.20(9)°.

The crystal structure of **4b**∙(CH_3_)_2_SO shows the hydroxy group H-bonding to the DMSO molecule of crystallisation (Figure 9a). The distance *d* for this H-bond is 1.89 Å. The twist angle between plane P(1)/P(2)/C(1)/C(2) and ring C(3) > C(8) is 72.2(4)°. Atoms N(1) and Pt(1) lie 0.781(17) and 0.180(10) Å away from the P(1)/P(2)/C(1)/C(2) plane, respectively. The hinge angle across the P(1)–P(2) vector is 8.7(6)°. Molecules form 1D, weakly H-bonded, undulating chains in the *c* direction via the methylene H atoms on C(1) and C(2) to a single, coordinated chloride ligand in an adjacent molecule (Figure 9b). Selected hydrogen bonding parameters for **4b**∙(CH_3_)_2_SO are shown in Table 9.

For compound **4e**∙½CHCl_3_∙½CH_3_OH there are two independent Pt complexes, one CH_3_OH, and one CHCl_3_ in the asymmetric unit. Both Pt complexes form 1D chains aligned parallel to *b*, but these chains are different (Figure 10). The chain involving Pt(2) forms simple O–H∙∙∙Cl H-bonds with the adjacent molecules via the *para* hydroxy group with *d* = 2.39(4) Å. For the chain involving the Pt(1)-containing molecules, the intermolecular H-bond has an inserted methanol molecule. The distances, *d*, are 2.32(5) and 1.82 Å for H(3)∙∙∙Cl(2) and H(1A)∙∙∙O(3), respectively. Atoms N(1)/N(2) and Pt(1)/Pt(2) lie 0.765(9)/0.798(9) and 0.424(5)/0.364(5) Å away from the P(1)/P(2)/C(1)/C(2) or P(3)/P(4)/C(34)/C(35) planes, respectively. So, as in **6**∙(CH_3_)_2_SO, the core 6-membered Pt–P–C–N–C–P rings adopt more chair-shaped conformations. The hinge angles across the P(1)–P(2)/P(3)–P(4) vectors are 13.44(16)/12.47(16)°. The twist angles between planes P(1)/P(2)/C(1)/C(2) or P(3)/P(4)/C(34)/C(35) and rings C(3) > C(8) or C(36) > C(41) are 88.17(19)/54.62(15)°. So, while the other geometric parameters are similar between the two molecules, this twist angle is significantly different.

In **5c**, the amide and ring atoms from C(4) > C(11) are disordered over two sets of almost equally occupied positions. The disorder highlights two or more chain-forming possibilities for this structure, analogous to that observed in in **4e**∙½CHCl_3_∙½CH_3_OH, with one possibility being simple (hydroxyl)O–H∙∙∙O(amide) links (Figure 11a), while the other, shown in Figure 11b, shows an alternative, water-inserted linkage. There is also likely to be some alternation of these motifs, given the random disorder and approx. 25% occupancy observed for water atom O(3). Unlike almost all of the other structures herein, the core 6-membered Pt–P–C–N–C–P ring adopts a conformation with atoms Pt(1)/P(1)/P(1)/C(2) being in a plane and atoms C(1) and N(2) being 1.021(6) and 1.237(6) Å, respectively, away from that plane. There is no C=O∙∙∙HN intermolecular H-bonding observed between molecules. Instead, the amide N*H* forms a bifurcated H-bond with the two neighbouring acceptor atoms N(1) and the *ortho* hydroxyl O(2) with *d* = 2.37 and 2.28 Å, respectively, while *d* = 2.89 Å for H(2)∙∙∙O(1A). 

In the second motif, adjacent molecules have an inserted water molecule in the H-bond pattern (Figure 11b). The amide N*H* again forms a bifurcated H-bond with the two neighbouring acceptor atoms N(1) and O(2X) with *d* = 2.14 and 2.25 Å, respectively, while *d* = 2.89 Å for H(2X)∙∙∙O(3), which is a little long, and *d* for O(3)∙∙∙O(1XA) = 2.21(3) Å, which is rather short. The distance *d* from water oxygen O(3) to O(1A), however, is entirely reasonable for an H-bond at 2.74 Å, suggesting a predominantly alternating pattern between the two disorder options is most likely.

Complex **5a**∙½Et_2_O was crystallised from a diethyl ether solution, including half a solvent molecule per complex molecule in the crystal lattice. There are two Pt complexes and two, half-occupied, Et_2_O solvent molecules of crystallisation in the asymmetric unit. The packing adopted by this second complex with an *ortho* hydroxyl group is very different to **5c** (Figure 12). Here there is no intramolecular N–H∙∙∙N H-bond, instead the ortho hydroxyl forms an intramolecular H-bond with the amide oxygen with *d* = 1.80 and 1.77(4) Å in the molecules containing Pt(1) and Pt(2), respectively. This does leave the two unique amide N*H* atoms free to form intermolecular interactions, which they do via highly asymmetric, bifurcated H-bonds with the coordinated chloride ligands on adjacent Pt complexes. From H(2) *d* = 2.60(11) and 2.95(13) Å to Cl(3) and Cl(4), respectively, while *d* = 2.52(7) and 3.12(15) Å from H(4) to Cl(1A) and Cl(2A), respectively. N(1)/N(3) and Pt(1)/Pt(2) lie 0.771(13)/0.781(14) and 0.349(8)/0.346(8) Å out of the planes P(1)/P(2)/C(1)/C(2) and P(3)/P(4)/C(37)/C(38), respectively. The twist angle between planes P(1)/P(2)/C(1)/C(2) and P(3)/P(4)/C(37)/C(38) relative to the rings C(5) > C(10) and C(41) > C(46) are 51.3(5) and 51.71(4)°, respectively. Hinge angles across P(1)–P(2) and P(3)–P(4) are 12.3(5) and 12.0(4)°, respectively. Differences between the two systems involving *ortho* hydroxyl groups are the position of the methyl ring substituent in the *meta* or *para* position, and the co-crystallised solvent being a small amount of water or Et_2_O. Either, or both of these differences might account for the different intra- and intermolecular packing motifs observed. Selected hydrogen bonding parameters for **5a**∙½Et_2_O are shown in Table 9.

Molecules of **5d**∙Et_2_O lie on a mirror plane, passing through Pt(1), between pairs of P and Cl atoms, and including the atoms from N(1) to the terminal hydroxy-substituted ring. Again, here the amide N*H* is involved in the 1D chain propagation (Figure 13), forming a symmetrical bifurcated H-bond with the two coordinated chloride ligands on the adjacent molecule with *d* = 2.66(15) Å. Supporting this is an additional (Ar)C–H(5)∙∙∙Pt(1) interaction at 2.78 Å. The twist angle between the P(1)/P(1A)/C(1)/C(1A) plane and the ring C(4) > C(9) = 90° due to the imposed crystallographic symmetry. The hinge angle at P(1)–P(1A) = 29.5(5)°. Atoms N(1) and Pt(1) lie 0.79(2) and 0.782(14) Å away from the P(1)/P(2)/C(1)/C(2) plane, respectively. So, this is the most chair shaped core Pt–P–C–N–C–P 6-membered ring. The *meta* hydroxyl group is not involved in the chain propagating intermolecular interactions and points into a cleft between a pair of Ph rings. It does not make an H-bond with the solvent of crystallisation.

For compound **5b**, the *para* position of the hydroxyl group facilitates 1D chain formation, forming an H-bond with one of the chloride ligands on an adjacent molecule with *d* = 2.09(6) Å (Figure 14). The amide N*H* here forms the familiar, but not universal, H-bond with the amine N(1) with *d* = 2.29(5) Å. The twist angle between the P(1)/P(2)/C(1)/C(2) plane and the ring C(5) > C(10) = 68.39(12)°. The hinge angle at P(1)–P(1A) = 4.95(10)°. Atoms N(1) and Pt(1) lie 0.810(4) and 0.164(3) Å away from the P(1)/P(2)/C(1)/C(2) plane, respectively.

## 3. Conclusions

In summary, we have shown that the position of the OH/CH_3_ groups with respect to the N-arene, the inclusion of an amide spacer, and the solvent used in the crystallisation can dictate the solid-state packing behaviour of both non coordinated and *cis*-PtCl_2_ bound diphosphine ligands. Unsurprisingly, the use of highly polar solvents (DMSO, CH_3_OH) in this study has been shown to play an important role in disrupting packing behaviour. Our work reinforces the importance of substituent effects, not only those commonly associated with −PR_2_ groups which may be alkyl or aryl based [37,38], but also those functional moieties positioned on the arene group of the central tertiary amine.

## 4. Materials and Methods

### 4.1. General Procedures

The synthesis of ligands **1a**–**e**, **2a**–**g**, and **3** were undertaken using standard Schlenk-line techniques and an inert nitrogen atmosphere. Ph_2_PCH_2_OH was prepared according to a known procedure [39]. All coordination reactions were carried out in air, using reagent grade quality solvents. The compound PtCl_2_(η^4^-cod) (cod = cycloocta-1,5-diene) was prepared according to a known procedure [40]. All other chemicals were obtained from commercial sources and used directly without further purification

### 4.2. Instrumentation

Infrared spectra were recorded as KBr pellets on a Perkin-Elmer Spectrum 100S (4000–250 cm^−1^ range) Fourier-Transform spectrometer. ^1^H NMR spectra (400 MHz) were recorded on a Bruker DPX-400 spectrometer with chemical shifts (δ) in ppm to high frequency of Si(CH_3_)_4_ and coupling constants (*J*) in Hz. ^31^P{^1^H} NMR (162 MHz) spectra were recorded on a Bruker DPX-400 spectrometer with chemical shifts (δ) in ppm to high frequency of 85% H_3_PO_4_. NMR spectra were measured in CDCl_3_ or (CD_3_)_2_SO at 298 K. Elemental analyses (Perkin-Elmer 2400 CHN Elemental Analyser) were performed by the Loughborough University Analytical Service within the Department of Chemistry. 

### 4.3. Preparation of Ligands ***1a***–***e***, ***2a***–***g***, and ***3***

The following general procedure was used for the synthesis of **1a**–**e**, **2a**–**g**, and **3**. A mixture of Ph_2_PCH_2_OH (2 equiv.) and the appropriate amine (1 equiv.) in CH_3_OH (20 mL) was stirred under N_2_ for 24 h. The volume of the solution was evaporated to ca. 2–3 mL, under reduced pressure, to afford the desired ligands which were collected by suction filtration (except **2a**–**c**) and dried *in vacuo*. Isolated yields in range 38–97%. Characterising details are given in Table 1.

### 4.4. Preparation of cis-Dichloroplatinum(II) Phosphine Complexes ***4a***–***e***, ***5a***–***g***, and ***6***

The following general procedure was used for the synthesis of **4a**–**e**, **5a**–**g**, and **6**. To a solution of PtCl_2_(η^4^-cod) (1 equiv.) in CH_2_Cl_2_ (5 mL) was added a solution of the appropriate ligand (1 equiv.) in CH_2_Cl_2_ (5 mL). The colourless (or pale yellow) solution was stirred for 30 min at r.t., evaporated to ca. 2–3 mL under reduced pressure, and diethyl ether (10 mL) added. The solids were collected by suction filtration and dried *in vacuo*. Isolated yields in range 73–99%. Characterising details are given in Table 4.

### 4.5. Single Crystal X-ray Crystallography

Suitable crystals of **1a**, **1b**∙CH_3_OH, **2f**·CH_3_OH, and **3** were obtained by slow evaporation of a CH_3_OH solution whereas **2g** was obtained by vapour diffusion of Et_2_O into a CDCl_3_/CH_3_OH solution. Crystals of **4b**∙(CH_3_)_2_SO, **5a**∙½Et_2_O, **5b**, and **5c**∙¼H_2_O were obtained by slow diffusion of Et_2_O into a CDCl_3_/(CH_3_)_2_SO/CH_3_OH solution. Slow diffusion of hexanes [for **6**∙(CH_3_)_2_SO] into a CDCl_3_/(CH_3_)_2_SO solution or vapour diffusion of Et_2_O into a CHCl_3_/(CH_3_)_2_SO/CH_3_OH [for **4c**∙CHCl_3_, **4e**∙½CHCl_3_∙½CH_3_OH) or CH_2_Cl_2_/CH_3_OH (for **5d**∙Et_2_O)]. Slow evaporation of a CH_2_Cl_2_/Et_2_O/hexanes solution gave suitable crystals of **4d**∙½Et_2_O. Table 2, Table 5 and Table 6 summarise the key data collection and structure refinement parameters. Diffraction data for compounds **1a**, **1b**∙CH_3_OH, **2f**∙CH_3_OH **3**, **4b**∙(CH_3_)_2_SO, **4c**∙CHCl_3_, **4d 4e**∙½CHCl_3_∙½CH_3_OH, **5d**∙Et_2_O, and **6**∙(CH_3_)_2_SO, were collected using a Bruker or Bruker-Nonius APEX 2 CCD diffractometer using graphite-monochromated Mo-K_α_ radiation. Data for compounds **5b** and **5c**∙¼H_2_O, were collected using a Bruker APEX 2 CCD diffractometer using synchrotron radiation at Daresbury SRS Station 9.8 or 16.2 SMX for **5a**·½Et_2_O. Data for compound **2g** was collected using a Bruker SMART 1000 CCD diffractometer using graphite-monochromated Mo-K_α_ radiation. All structures were solved by direct methods [except structures **4b**∙(CH_3_)_2_SO, **5a**∙½Et_2_O, and **5b** which were solved using Patterson synthesis] and refined by full-matrix least-squares methods on *F*^2^. All C*H* atoms were placed in geometrically calculated positions and were refined using a riding model (aryl C–H 0.95 Å, methyl C–H 0.98 Å, methylene C–H 0.99 Å. Where data quality allowed, O*H* and N*H* atom coordinates and *U*_iso_ were freely refined, or refined with mild geometrical restraints; otherwise, they were placed geometrically with O/N–H = 0.84 Å. *U*_iso_(*H*) values were set to be 1.2 times *U*_eq_ of the carrier atom for aryl C*H* and N*H*, and 1.5 times *U*_eq_ of the carrier atom for O*H* and C*H*_3_. Throughout the text and tabulated data, where H atom geometry does not include a SU, the coordinates were constrained. Unless stated, all structural determinations proceeded without the need for restraints or disorder modelling. Where disorder was modelled it was supported with appropriate geometrical and *U* value restraints. In **1b**∙CH_3_OH, the methanol was modelled as disordered over two equally occupied sets of positions. In **2f**·CH_3_OH the methanol was modelled using the Platon Squeeze procedure [41]. Compound **3** was found to contain a disordered methanol and was modelled over two sets of positions, each at half weight. In **4d**·½Et_2_O, atoms C(1) > C(7) and N(1) were modelled with *U* value restraints. The Et_2_O was modelled using Platon Squeeze due to significant disorder. In **4e**∙½CHCl_3_∙½CH_3_OH the chloroform molecule was modelled over two sets of positions with major occupancy 57.1(7)% Restraints were applied to that molecule and also ring C(55) > C(60). In **5a**·½Et_2_O three Ph rings were modelled as disordered over two sets of positions with occupancies close to 50%. Restraints were applied to these rings and also the two half-occupancy Et_2_O solvent molecules of crystallisation. In **5c**∙¼H_2_O, atoms Cl(1) and C(3) > C(11), O(1), O(2) and N(1) were modelled as split over two sets of positions with major occupancy 56(4) and 50.9(6)%, respectively and restraints were applied. In **5d**∙Et_2_O the Et_2_O was modelled as a diffuse area of electron density by the Platon Squeeze procedure and restraints were applied to atoms C(1) > C(10), C(11) > C(22) and N(2) O(2). In **6**∙(CH_3_)_2_SO the DMSO was modelled with restraints as disordered over two sets of positions with major component 71.0(5)% and with C(33) coincident for both components Programs used during data collection, refinement and production of graphics were Bruker SMART, Bruker APEX 2, SAINT, SHELXTL, COLLECT, DENZO and local programs [41,42,43,44,45,46,47,48,49,50,51]. CCDC 2101643-2101656 contain the supplementary crystallographic data for this paper. These data can be obtained free of charge from The Cambridge Crystallographic Data Centre via www.ccdc.cam.ac.uk/structures (accessed on 3 November 2021).

## Data Availability

Not applicable.

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
