# Peer review of "Low-Dimensional Architectures in Isomeric cis-PtCl2{Ph2PCH2N(Ar)CH2PPh2} Complexes Using Regioselective-N(Aryl)-Group Manipulation"

_molecules, 2021, doi:10.3390/molecules26226809_

Round 1

Reviewer 1 Report

This manuscript describes the supramolecular behaviour of two series of isomeric, phenol-substituted, ami-nomethylphosphines as free ligands and part of the complexes. This work revealed that the position of the OH/CH3 groups with respect to the N-arene and the solvent used in the crystallisation can dictate the supramolecular behaviour of both non coordinated and cis-PtCl2 bound diphosphine ligands. These findings may be useful for developing new supramolecular systems. The manuscript is well written and presented. Furthermore, the crystal structures of the complexes were well characterized and discussed.

This manuscript will be suitable for submission as a full article after revision about following points.

(1) Please check the reference part. The numbering is duplicated.

 For example;  1. 1. Lehn, J.-M. Supramolecula....

(2) I think that 13C NMR spectra are necessary to identify the synthesized organic compounds (ligands). Is it not possible to measure it? 

Author Response

We note that our original m/s has been changed (Editorial office) and hence the duplication in reference numbers. The revised manuscript has one set of reference numbers.
Our studies in this field began over 15 years ago and we have previously published other aspects of this work (see refs 10c-10f). In our work, inclusive of the studies described here, we are confident of both purity and structure assignment from the combination of NMR [1H and 31P{1H}] alone, FT−IR, mass spectrometry, elemental analysis and single crystal X-ray crystallography.

Reviewer 2 Report

I recommend the following consideration for the publication.

  1. I suggest to use “2g” instead of g for the compound in the abstract section.
  2. In Scheme 1, please check the extra double down arrows and the missing reagents. Here compound 3 is drawn but the preparation scheme is missing which can be represented like 1a-e.
  3. For the preparation of 2a-g, authors mentioned using primary amine. I got confusion in scheme 1 and explanation section (text) that whether authors used amide-functionalised primary amine or the similar one used to prepare 1a-e.
  4. I was also wondered about the yield of product around 99%.
  5. I suggest to check the space group for the compounds in Table 2.
  6. Table 3 looks mess; I suggest to redraw table for the clarity.
  7. The structure description for complexes 4a and 5e are missing. Please check it.
  8. Please check the error in the last paragraph in page 12, “…. Figure 4.d. ½ Et2O….”
  9. I suggest to explain the crystal structural description in sequence. Due to the amide functional group and solvent nature, there may be supramolecular structure which are missing under the title of Low-Dimensional Architectures in complexes.
  10. I suggest to mention figure 11a and 11b in diagram.
  11. There are missing of structural description of 4a and 5e in the manuscript.
  12. The beauty of the manuscript is to syntheses of two series of isomeric ligands and complexes using them. Authors synthesized 2f and 2g for the comparative purposes of non-methyl substituted diphosphines with the substituted one. However, there is no explanation about this effect compare to the other for the complex formation as well as supramolecular structure diversity.
  13. In the conclusion section, I suggest to explain about the finding with the reasons not just like the report.
  14. In the low (0) dimensional coordinated complexes, as simulated and as synthesized PXRD patterns may be shifted. However, for the purity of complexes, I recommend to show the simulated and synthesized PXRD patterns.
  15. I recommend to submit the CheckCIF file and CIF file for the compounds.

Author Response

1. We have amended this accordingly.
2. Scheme 1 has been amended to include the reagents and compound 3 represented like 1a-e. We have also revised Chart 2 accordingly.
3. The text has been modified to improve the clarity for the preparation of the primary alkylamines used to synthesise 2a-g.
4. The isolated yield of 99% for compound 5d is correct.
5. Our X-ray crystallographer, Dr Elsegood, has confirmed the space groups for the compounds in Table 2 are all indeed correct.
6. We note that our original Table 3 (word file) is fine. The Editorial office have changed the Table hence it appearing as a “mess”. In general, it should also be added all the other tables appear incorrect and/or split over two pages (please see original/revised tables that we have submitted).
7. No suitable crystals of compounds 4a and 5e could be obtained and hence no single crystal X-ray data is presented in this manuscript.
8. In our original submitted manuscript (pg 13), the text correctly reads “For 4d∙½Et2O (Figure 8) a molecule of badly disordered…” The Editorial office have changed this hence the error in this last paragraph.
9. Our reasoning for the order presented in the manuscript reflects the progression in packing arrangements on going from compounds 5c to 5a to 5d to 5b.
10. In our original submitted manuscript (pg 17), Figures 11(a) and (b) are clearly shown. The Editorial office have changed this hence the omission in the version being reviewed (pg 15).
11. See point 7. above (as this comment is a duplication already covered).
12. We were delighted about obtaining crystals of 2f and 2g and, although we have prepared the corresponding platinum(II) complexes 5f and 5g, which are now included in the revised manuscript, we were unable to obtain any X-ray quality crystals.
13. We have amended the Conclusion slightly. Without extensive further studies, deep meaningful reasons to explain the range of packing effects observed is difficult.
14. No PXRD measurements were collected for any of the compounds described here. We use elemental analysis as a means of providing bulk material characterisation and use this, in conjunction with our single crystal X-ray data.
15. CIF files have been submitted and provided by our X-ray crystallographer.

Reviewer 3 Report

  1. The work comprises the rigorous description of crystal structures of novel aminomethylphosphines and their complexes with PtCl2. However, the supramolecular chemistry features of these compounds are not well pronounced. it seems that the authors consider the presence of intermolecular hydrogen bonds as an ideal sign of the supramolecular nature of the compound. However, it is only the first sign, but not the last. Supramolecular interactions should lead to change of physical or chemical properties of compounds. But all "supramolecular compounds" are only solvates, the structures of which are very abundant in CCDC. In this work a reader can not compare properties of compounds, besides of RSA data, with supramolecular interactions and without them. So, I would recommend to add more conclusions (or forecasts) about supramolecular features of new compounds or change the article focus to crystallographic.
  2. According to rules "Reports on previously undescribed organic compounds should include, as supplementary data, 1H, 13C and/or other key heteronuclear or 2D NMR spectra, together with High Resolution Mass Spectrometry (HRMS) or elemental analysis."
    There are no 13С data for new compounds expect 1b.
  3. Page 5 "Figure’s 1-5" should be " Figures 1-5"
  4. Page 12. " Figure 4. d∙½ Et2O (Figure 8)" should be "Compound 4d..."?
  5. Page 15. Does the bond between two oxygens O(3) and O(1XA) really exist? To which atoms the hydrogens of water oxygen O(3) are bound?

Author Response

1. We note the comments made by this Reviewer and have replaced, in various cases, “supramolecular” with “solid-state packing” to avoid any confusion.
2. Our studies in this field began over 15 years ago and we have previously published other aspects of this work (see refs 10c-10f). In our work, inclusive of the studies described here, we are confident of both purity and structure assignment from the combination of NMR [1H and 31P{1H}] alone, FT−IR, mass spectrometry, elemental analysis and single crystal X-ray crystallography.
3. We have amended this accordingly.
4. In our original submitted manuscript (pg 13), the text correctly reads “For 4d∙½Et2O (Figure 8) a molecule of badly disordered…” The Editorial office have changed this hence the error in this last paragraph.
5. Taking on comments from our X-ray crystallographer, we have now modified the text in the revised manuscript (pg 17) accordingly.

Round 2

Reviewer 3 Report

In the corrected version of the manuscript I see "supramolecular" in the Abstract, which is not supported by Conclusion.

Also, in page 7 the phrase "The supramolecular synthons observed in the solid state for these highly modular ligands...." presents. If "these ligands" are compounds 1a, 1b∙CH3OH, 2f∙CH3OH, 2g, and 3, I do not agree with this phrase because I can see "supramolecular" interactions only in 1b∙CH3OH but not in other substances. I would propose to change this phrase or omit any "supramolecular" at all.
